



# Quantifying overlapping and differing information of global precipitation for GCM forecasts and El Niño–Southern Oscillation

Tongtiegang Zhao[1,2], Haoling Chen[1], Yu Tian[3] and Xiaohong Chen[1,2]

[1] Center of Water Resources and Environment, School of Civil Engineering, Sun Yat-Sen University, Guangzhou, China
[2] Southern Marine Science and Engineering Guangdong Laboratory, Zhuhai, China
[3] Department of Water Resources, Institute of Water Resources and Hydropower Research of China, Beijing, China

*Correspondence to*: Tongtiegang Zhao (zhaottg@mail.sysu.edu.cn) and Xiaohong Chen (eescxh@mail.sysu.edu.cn)

**Abstract.** While El Niño–Southern Oscillation (ENSO) teleconnection has long been used in statistical hydroclimatic forecasting, global climate models (GCMs) provide increasingly available dynamical precipitation forecasts for hydrological
modelling and water resources management. It is not yet known to what extent dynamical GCM forecasts provide new information compared to statistical teleconnection. This paper develops a novel Set Operations of Coefficients of Determination (SOCD) method to explicitly quantify the overlapping and differing information for GCM forecasts and ENSO teleconnection. Specifically, the intersection operation of the coefficient of determination derives the overlapping information for GCM forecasts and Niño3.4 index, and then the difference operation determines the differing information in GCM forecasts
(Niño3.4 index) from Niño3.4 index (GCM forecasts). A case study is devised for the Coupled Forecast System model version 2 seasonal forecasts of global precipitation in December-January-February. The results show that the overlapping information for GCM forecasts and Niño3.4 index is significant for 34.94% of global land grid cells, the differing information in GCM forecasts from Niño3.4 index is significant for 31.18% of grid cells and the differing information in Niño3.4 index from GCM forecasts is significant for 11.37% of grid cells. These results confirm the effectiveness of GCMs in capturing the ENSO-
related variability of global precipitation and also illustrate where there is room for improvements of GCM forecasts. Furthermore, the bootstrapping-based significance tests of the three types of information facilitate in total eight patterns to disentangle the close but divergent association of GCM forecast correlation skill with ENSO teleconnection.

## 1 Introduction

Precipitation is one of the most important climate forcings of hydrological processes at catchment, regional and continental
scales (Wada and Bierkens, 2014; Li et al., 2017; Touma et al., 2018; Pechlivanidis et al., 2020; Chen et al., 2022). Performing hydrological forecasting into the future, the uncertainty generally arises from catchment initial conditions and future climate forcings (Wood and Lettenmaier, 2006; Yossef et al., 2013; Yuan et al., 2016; Huang et al., 2020; Chen et al., 2021). In a short lead time up to about one month, initial conditions tend to outweigh climate forcings; at longer lead times, climate forcings become a more important contributor (Li et al., 2009). Therefore, besides remote sensing-based estimations of initial conditions
of snow cover, soil moisture and groundwater storage (Mei et al., 2020; Xu et al., 2020; Zhang et al., 2021), efforts have been





devoted to developing sub-seasonal to seasonal hydroclimatic forecasts of temperature and precipitation (Schepen et al., 2016; Bennett et al., 2017; Cash et al., 2017; Pendergrass et al., 2020). While temperature forecasts have been improved substantially in the past decades, the generation of skilful precipitation forecasts remains a challenging task (Becker et al., 2020).

Climate indices, in particular El Niño–Southern Oscillation (ENSO, Mason and Goddard, 2001), have been conventionally
used in hydroclimatic forecasting (Hamlet and Lettenmaier, 1999; Hidalgo and Dracup, 2003; Peel et al., 2004; Yang et al., 2018; Lim et al., 2021). Teleconnections with climate indices generally reflect slowly varying and recurrent components, for example sea surface temperature (SST), of atmospheric circulations that link climate anomalies over large distances in both the tropics and extratropics (Webster and Yang, 1992; Mason and Goddard, 2001). There are generally three types of teleconnection patterns: meridional dipole, wave and hybrid (Kim et al., 2021). As one of the most remarkable teleconnections,
ENSO affects the global climate through eastward propagating Kelvin waves, westward propagating Rossby waves and Walker circulations that span the tropical Pacific, Indian and Atlantic Oceans (Yang et al., 2018). For regions exhibiting teleconnection patterns, various models have been developed for the purpose of hydroclimatic forecasting, including historical resampling methods (Hamlet and Lettenmaier, 1999; Wood and Lettenmaier, 2006; Lim et al., 2021), statistical (Bayesian) methods (Hidalgo and Dracup, 2003; Emerton et al., 2017; Strazzo et al., 2019) and machine learning methods (Xu et al., 2020; Li et
al., 2021).

Major climate centers develop global climate models (GCMs) to generate operational forecasts of global climate (Saha et al., 2014; Khan et al., 2017; Johnson et al., 2019; Delworth et al., 2020; Lin et al., 2020). For example, the United States National Centers for Environmental Prediction (NCEP) runs the Coupled Forecast System model version 2 (CFSv2, Saha et al., 2014) and the European Centre for Medium-Range Weather Forecasts operates the fifth-generation seasonal forecast system (SEAS5,
Johnson et al., 2019). In contrast to teleconnections that are generally "statistical", GCM forecasts are "dynamical" in that GCMs assimilate observational information to reduce initial state uncertainty and couple atmosphere, land, ocean and sea ice modules to formulate complex interactions among different components of the earth system (Bauer et al., 2015; Corti et al., 2015; Li et al., 2017; Johnson et al., 2019; Strazzo et al., 2019). Previous studies found that GCM forecasts tend to be skilful in regions subject to prominent ENSO teleconnection (Johnson et al., 2019; Delworth et al., 2020; Zhao et al., 2021) and also
highlighted that GCM forecasts can be skilful in some extratropical regions where there is limited ENSO teleconnection (Li et al., 2017; Johnson et al., 2019; Zhao et al., 2021).

Conventional ENSO-based statistical forecasts and emerging GCM dynamical forecasts provide valuable information for hydrological modelling and water resources management (Wood and Lettenmaier, 2006; Bauer et al., 2015; Emerton et al., 2017; Strazzo et al., 2019; Delworth et al., 2020). As the two types of forecasts are independently generated, it would not be
surprising if statistical forecasts outperform dynamical forecasts in some regions but underperform in some other regions. However, it is not yet known to what extent their information overlaps or differs. Small overlap and large difference highlight that GCM forecasts do offer new information comparing to ENSO teleconnection, while large overlap and small difference imply that GCM forecasts might not provide additional information. Zhao et al. (2021) investigated the overlapping information by developing a set-theory-based approach to attributing GCM forecast correlation skill to ENSO teleconnection.





In this paper, we build a Set Operations of Coefficients of Determination (SOCD) method upon Zhao et al. (2021) to furthermore account for the differing information. As will be demonstrated through the methods and results, besides the overlapping information, there exist two types of differing information, i.e., the differing information in GCM forecasts from ENSO and the differing information in ENSO from GCM forecasts. The three types of information facilitate eight patterns to disentangle the close but divergent association of GCM correlation skill with ENSO teleconnection.


## 2 Data description

GCM precipitation forecasts are generally five-dimensional data (Kirtman et al., 2014; Delworth et al., 2020; Lin et al., 2020). Taking the NCEP-CFSv2 forecasts (Saha et al., 2014) as an example, the five dimensions are: 1) forecast start time $s$, which represents the time at which forecasts are generated, is marked by the number of months since January 1960; 2) lead time $l$,

which represents the months ahead the start time, ranges from 0 to 9; 3) ensemble member $n$, which is meant to explicitly account for forecast uncertainty, ranges from 1 to 24, i.e., 24 ensemble members in total; 4) latitude $y$; and 5) longitude $x$. GCM forecasts are therefore formulated as:

$$F = \left[ f_{s,l,n,y,x} \right],$$  (1)

where $f$ represents individual forecast value under the five dimensions and all the forecast values form a dataset $F$.

The observed precipitation corresponding to the forecasts has three dimensions:

$$O = \left[ o_{t,y,x} \right] \quad (t = s + l),$$  (2)

in which $o$ represents individual observation value and $O$ the dataset of observations. The three dimensions are target time $t$, latitude $y$ and longitude $x$. It is important to note that target time $t$ is mathematically the sum of start time $s$ and lead time $l$ in aligning observations with forecasts.

*Niño*3.4 that indicates the SST of the East Central Tropical Pacific (5ºN–5ºS, 170º–120ºW) is one of the most popular indicators of the status of ENSO (Hamlet and Lettenmaier, 1999; Barnston et al., 2012; Saha et al., 2014; Emerton et al., 2017; Lin et al.,

85   2020):

$$Niño3.4=[niño3.4_t],$$  (3)

in which there is only one dimension, i.e., time $t$, for *Niño*3.4.

$F$, $O$ and *Niño*3.4 shown in Eqs. (1) to (3) lay the basis for the analysis of overlapping and differing information in this paper. In the North American Multi-Model Ensemble (NMME) experiment (Kirtman et al., 2014), CFSv2 retrospective forecasts that range from 1982 to 2010 have been temporally aggregated to monthly and spatially regridded to a 1.0º×1.0º resolution. In the

meantime, the daily Unified Rain-gauge Database of the Climate Prediction Center (CPC-URD, Chen et al., 2008) precipitation observations over land have also been aggregated and regridded by the NMME. In the analysis, both CFSv2 forecasts and CPC-URD observations are obtained from the International Research Institute of the Columbia University





(https://iridl.ldeo.columbia.edu/SOURCES/.Models/.NMME/). Monthly *Niño*3.4 is also obtained from the CPC (https://www.cpc.ncep.noaa.gov/data/indices/).


## 3 Methods

### 3.1 Consideration of seasonality

Precipitation worldwide exhibits seasonality, e.g., wet and dry seasons of monsoonal precipitation (Webster and Yang, 1992; Chen et al., 2020; Zhao et al., 2021). As a result, the predictive performance of GCM forecasts varies across different seasons
(Kirtman et al., 2014; Saha et al., 2014; Bauer et al., 2015) and ENSO teleconnection also exhibits seasonal variabilities (Peel et al., 2004; Timmermann et al., 2018). By fixing the target season, lead time $l$ would be determined by start time $s$. Taking December-January-February (DJF) (Webster and Yang, 1992; Mason and Goddard, 2001) for an example, forecasts generated at the start of December are at 0-month lead time, forecasts at the start of November are at 1-month lead time, and so on. Considering seasonality, the start time $s$ in Eq. (1) is re-formulated by month $m$ and year $k$, e.g., December 1982, December
1983, …, and December 2010. By fixing the target season and specifying the start month, GCM forecasts are then extracted from $F$:

$$F_{Dec \rightarrow DJF, y, x} = \left[ \overline{f}_k \right].$$

(4)

The five dimensions of $F$ (Eq. 1) are handled as follows: 1) start time $s$ and lead time $l$ are replaced by $Dec \rightarrow DJF$ and then represented by $k$, i.e., aggregating monthly forecasts into seasonal and pooling forecasts across different years; 2) ensemble member $n$ is eliminated by taking the mean value ($\overline{f}$), i.e., the ensemble mean, of all ensemble members (e.g., Saha et al.,
2014; Khan et al., 2017; Lin et al., 2020) in the analysis; and 3) latitude $y$ and longitude $x$ are pre-specified for the extraction of forecasts.

The observations corresponding to the forecasts in Eq. (4) are extracted from the dataset $O$ (Eq. 2):

$$O_{DJF, y, x} = \left[ o_k \right].$$

(5)

In Eq. (5) is observed precipitation in the target season (DJF) across multiple years at the selected grid cell ($y$, $x$). Similar to forecasts, monthly observations are aggregated into seasonal.
Furthermore, the *Niño*3.4 index in the same season as observed precipitation is obtained:

$Niño3.4_{DJF} = [niño3.4_k]$. (6)

In Eq. (6) is the concurrent *Niño*3.4 of the target season (DJF) across multiple years.





### 3.2 Quantification of information in forecasts and Niño3.4

The coefficient of determination ($R^2$) is effective in quantifying the proportion of the variance of dependent variable explained
by a regression model that is built upon some independent variable(s) (Pham, 2006). In this paper, the dependent variable is observed seasonal precipitation (Eq. 5). The candidate independent variables are GCM precipitation forecasts (Eq. 4) and *Niño*3.4 index (Eq. 6). Three classic simple linear regression models are set up to account for the information of observations in forecast ensemble mean and *Niño*3.4 index.

The first model regresses observed seasonal precipitation $o$ against ensemble mean $\overline{f}$ of GCM precipitation forecasts:

$$o_k = \alpha_1 + \beta_1 \overline{f}_k + \varepsilon_{1,k} \Rightarrow R^2(o \sim \overline{f}) = 1 - \frac{\sum_k \varepsilon_{1,k}^2}{\sum_k (o_k - \overline{o})^2}, \tag{7}$$

in which $\alpha_1$ and $\beta_1$ are respectively the intercept and slope parameters of simple linear regression. The unexplained variance indicated by the sum of squared residual, i.e., $\sum_k \varepsilon_{1,k}^2$, is compared to the variance of observed precipitation $\sum_k (o_k - \overline{o})^2$. In this way, the proportion of variance explained by ensemble mean is quantified.

The second model regresses observed seasonal precipitation $o$ against *Niño*3.4 index:

$$o_k = \alpha_2 + \beta_2 ni\tilde{n}o3.4_k + \varepsilon_{2,k} \Rightarrow R^2(o \sim ni\tilde{n}o3.4) = 1 - \frac{\sum_k \varepsilon_{2,k}^2}{\sum_k (o_k - \overline{o})^2}, \tag{8}$$

in which $\alpha_2$, $\beta_2$ and $\varepsilon_{2,k}$ are respectively the intercept parameter, slope parameter and residual of regression. This
regression quantifies the proportion of variance of observed precipitation explained by *Niño*3.4.

The third model regresses observed seasonal precipitation $o$ against both ensemble mean $\overline{f}$ and *Niño*3.4 index:

$$o_k = \alpha_3 + \beta_{3,1} \overline{f}_k + \beta_{3,2} ni\tilde{n}o3.4_k + \varepsilon_{3,k}$$

$$\Rightarrow R^2(o \sim \overline{f} \cup ni\tilde{n}o3.4) = 1 - \frac{\sum_k \varepsilon_{3,k}^2}{\sum_k (o_k - \overline{o})^2}, \tag{9}$$

in which $\alpha_3$, $\beta_{3,1}$, $\beta_{3,2}$ and $\varepsilon_{2,k}$ are respectively the intercept parameter, slope parameter of ensemble mean, slope parameter of *Niño*3.4 and residual of regression. The proportion of the variance of observed precipitation explained by the union of ensemble mean and *Niño*3.4 index is therefore measured by this bi-variate regression.




### 3.3 Quantification of overlapping and differing information

As shown by Venn diagrams in Figure 1, the information of observed precipitation contained in forecast ensemble mean, *Niño*3.4 index and their union are respectively quantified by $R^2(o \sim \overline{f})$, $R^2(o \sim ni\tilde{n}o3.4)$ and $R^2(o \sim \overline{f} \cup ni\tilde{n}o3.4)$. Furthermore, the SOCD method performs the set operations of intersection and difference to quantify the overlapping and

differing information:

$$R^2(o \sim \overline{f})$$

$$R^2(o \sim ni\tilde{n}o3.4)$$

$$R^2(o \sim \overline{f} \cup ni\tilde{n}o3.4)$$

$$R^2(o \sim \overline{f}/ni\tilde{n}o3.4) \qquad R^2(o \sim \overline{f} \cap ni\tilde{n}o3.4) \qquad R^2(o \sim ni\tilde{n}o3.4/\overline{f})$$

**Figure 1: Venn diagram representation of the set operations of union, intersection and difference to quantify the overlapping and differing information. The different terms of information are measured by the classic coefficient of determination.**


1) The proportion of variance explained by ensemble mean but not by *Niño*3.4 index is derived by the difference operation:

$$R^2(o \sim \overline{f}/ni\tilde{n}o3.4) = R^2(o \sim \overline{f} \cup ni\tilde{n}o3.4) - R^2(o \sim ni\tilde{n}o3.4) \qquad (10)$$





In Eq. (10), $R^2(o \sim \overline{f}/ni\tilde{n}o3.4)$ measures the differing information of GCM forecasts on observed precipitation from *Niño*3.4 index.

2) The intersection operation derives the proportion of variance of seasonal precipitation explained by both ensemble mean and *Niño*3.4 index:

$$R^2(o \sim \overline{f} \cap ni\tilde{n}o3.4)$$
$$= R^2(o \sim \overline{f}) + R^2(o \sim ni\tilde{n}o3.4) - R^2(o \sim \overline{f} \cup ni\tilde{n}o3.4) \quad (11)$$

In Eq. (11), $R^2(o \sim \overline{f} \cap ni\tilde{n}o3.4)$ represents the overlapping information.

3) The proportion of variance explained by *Niño*3.4 index but not by ensemble mean is derived by the difference operation:

$$R^2(o \sim ni\tilde{n}o3.4/\overline{f}) = R^2(o \sim \overline{f} \cup ni\tilde{n}o3.4) - R^2(o \sim \overline{f}) \quad (12)$$

In Eq. (12), $R^2(o \sim ni\tilde{n}o3.4/\overline{f})$ represents the differing information of *Niño*3.4 index from GCM forecasts.

### 3.4 Eight patterns for overlapping and differing information

The significance of overlapping and differing information is tested by bootstrapping (Efron and Tibshirani, 1986). It is because these $R^2$ values are generated not directly by linear regressions but by the set operations of $R^2(o \sim \overline{f})$, $R^2(o \sim ni\tilde{n}o3.4)$ and $R^2(o \sim \overline{f} \cup ni\tilde{n}o3.4)$ in Eqs. (7), (8) and (9). As a result, they do not follow standard F-tests of significance. In bootstrapping, the null hypothesis is that the three variables under investigation, i.e., $o$, $\overline{f}$ and $ni\tilde{n}o3.4$, were fully independent from one another. Accordingly, by randomly permuting the order of samples in Eqs. (4), (5) and (6), the overlapping and differing information are re-calculated; one thousand such permutations formulate the respective reference distributions for these $R^2$ values under the null hypothesis. Comparing the $R^2$ values for the original samples respectively to their reference distributions, the p-values are obtained to tell how extreme the $R^2$ values for the original samples are.

As the null hypothesis is full independence, the $R^2$ values, which indicate the amount of information of dependent variable contained in independent variable(s) (Pham, 2006), are expected to be rather small. From this perspective, the larger the $R^2$ values for the original samples are, the more extreme they are and the less likely the null hypothesis holds. Therefore, instead of the popular two-tailed test (Pham, 2006), the one-tailed test is implemented for the significance of the $R^2$ values. Specifically, under the significance level of 0.10, the SOCD method pays attention to whether the $R^2$ value falls into the top 10% of the corresponding bootstrapping-derived reference distribution. The three terms of overlapping versus differing information each have two cases of significance, i.e., significant or non-significant. Accordingly, as shown in Table 1, the significance tests facilitate in total 8 (2*2*2) patterns.





**Table 1: Eight patterns of overlapping and differing information. In the first three columns, 1 and 0 respectively indicate the significant and non-significant cases.**

| $R^2(o \sim \overline{f} / ni\tilde{n}o3.4)$ | $R^2(o \sim \overline{f} \cap ni\tilde{n}o3.4)$ | $R^2(o \sim ni\tilde{n}o3.4/\overline{f})$ | Meaning |
|---|---|---|---|
| 0 | 0 | 0 | Neither overlapping information nor differing information is significant |
| 0 | 0 | 1 | Only the differing information in *Niño*3.4 index from GCM forecasts is significant |
| 0 | 1 | 0 | Only the overlapping information is significant |
| 0 | 1 | 1 | Both overlapping information and differing information in *Niño*3.4 index from GCM forecasts are significant |
| 1 | 0 | 0 | Only the differing information in GCM forecasts from *Niño*3.4 index is significant |
| 1 | 0 | 1 | Both differing information in GCM forecasts from *Niño*3.4 index and differing information in *Niño*3.4 index from GCM forecasts are significant, but the overlapping information is not significant |
| 1 | 1 | 0 | Both differing information in GCM forecasts from *Niño*3.4 index and overlapping information are significant |
| 1 | 1 | 1 | Differing information in GCM forecasts from *Niño*3.4 index, overlapping information and differing information in *Niño*3.4 index from GCM forecasts are all significant |


## 4 Results

### 4.1 Spatial plots of correlation skill and ENSO teleconnection

GCM forecast Correlation skill and ENSO teleconnection for DJF are respectively shown in the upper and lower parts of Figure 2. The correlation skill is mathematically the Pearson's correlation coefficient between GCM forecast ensemble mean and observed precipitation. In the upper part of Figure 2, it is observed that the correlation skill is higher than 0.3 in a substantial

number of grid cells around the world. This result indicates that ensemble mean can be indicative of observed precipitation, i.e., high values of ensemble mean coincide with high values of observed precipitation and vice versa (Saha et al., 2014; Yuan





et al., 2014; Cash et al., 2019). In the lower part of the Figure is ENSO teleconnection that is mathematically the Pearson's correlation coefficient between *Niño*3.4 index and observed precipitation. Both positive and negative ENSO teleconnections

are observed. For example, the teleconnection tends to be positive in southern North America, south-eastern South America, southern China and Eastern Africa, implying above-average precipitation in El Niño years but below-average precipitation in La Niña years; and it turns out to be negative in the northern part of South America, southern Africa as well as Southeast Asia, i.e., there can be below-average precipitation in El Niño years and above-average precipitation in La Niña years (Mason and Goddard, 2001; Emerton et al., 2017; Yang et al., 2018).


**Figure 2: Correlation skill and ENSO teleconnection for global precipitation in DJF. Correlation skill represents the correlation between precipitation observations and seasonal CFSv2 forecasts generated at the start of December; and ENSO teleconnection represents the correlation between precipitation observations and the concurrent Niño3.4 in DJF.**


**Figure 3: Spatial distribution of the eight SOCD-derived patterns (upper part) and Venn diagrams of overlapping and differing information for selected grid cells under the eight patterns (lower part). The eight grid cells A to H are marked in the spatial plots of correlation skill and ENSO teleconnection in Figure 2.**


Through the set operations and significance tests, the SOCD method derives eight patterns to characterize the overlapping and differing information for GCM forecast ensemble mean and $Niño3.4$ index. The spatial distribution of the patterns is shown in the upper part of Figure 3. In addition, eight grid cells are selected from Figure 2 to showcase the eight patterns: in the lower part of Figure 3, the coefficients of determination that play a critical part in the patterns are represented by the areas of Venn

diagrams. Grid cells under the pattern 000 are in grey: it can be observed that grey areas in the upper part of Figure 3 generally





correspond to areas of poor GCM correlation skill and limited ENSO teleconnection in Figure 2. Also, the corresponding Venn diagram in the lower part of Figure 3 suggests that little information of observed precipitation is contained in forecast and *Niño*3.4. In the meantime, it is highlighted that a considerable amount of grid cells around the world are colored. The corresponding Venn diagrams tend to be of large areas, highlighting that either the overlapping information or the differing

information is significant. From the upper and lower parts of Figure 2, it can be found that positive correlation skill corresponds to positive ENSO teleconnection in southern North America and Eastern Africa, and to negative teleconnection over the northern part of South America, southern Africa and Southeast Asia. In the meantime, from Figure 3 it can be seen that in these regions a considerable number of grid cells fall under the patterns 010, 110 and 011, indicating significant overlapping information.


### 4.2 Patterns of overlapping and differing information

The patterns derived by the SOCD method serve as a link between correlation skill and ENSO teleconnection. The pattern 010 that is concentrated on the overlapping information is shown in Figure 4. At the left-hand side of the figure are the results for grid cells under the pattern 010 (the results for the other grid cells are masked). The overlapping information is significant in

southern North America where positive correlation skill (upper left part of Figure 4) coincides with positive ENSO teleconnection (lower left part of Figure 4). It is also significant in southern Africa and northern South America where positive correlation skill and negative ENSO teleconnection coexist. As both correlation skill and ENSO teleconnection are mathematically the Pearson's correlation coefficient, they each can be classified into three cases, i.e., significantly positive (P), non-significant (ns) and significantly negative (N) (Pham, 2006). At the right-hand side of Figure 4, the Sankey diagram shows

that 18.95% of the global land grid cells exhibit pattern 010. For this pattern, 8.98% of grid cells exhibit significantly positive correlation skill, 9.85% non-significant correlation skill and 0.12% significantly negative correlation skill; 3.77% exhibiting significantly positive ENSO teleconnection, 10.92% non-significant ENSO teleconnection and 4.25% significantly negative ENSO teleconnection.





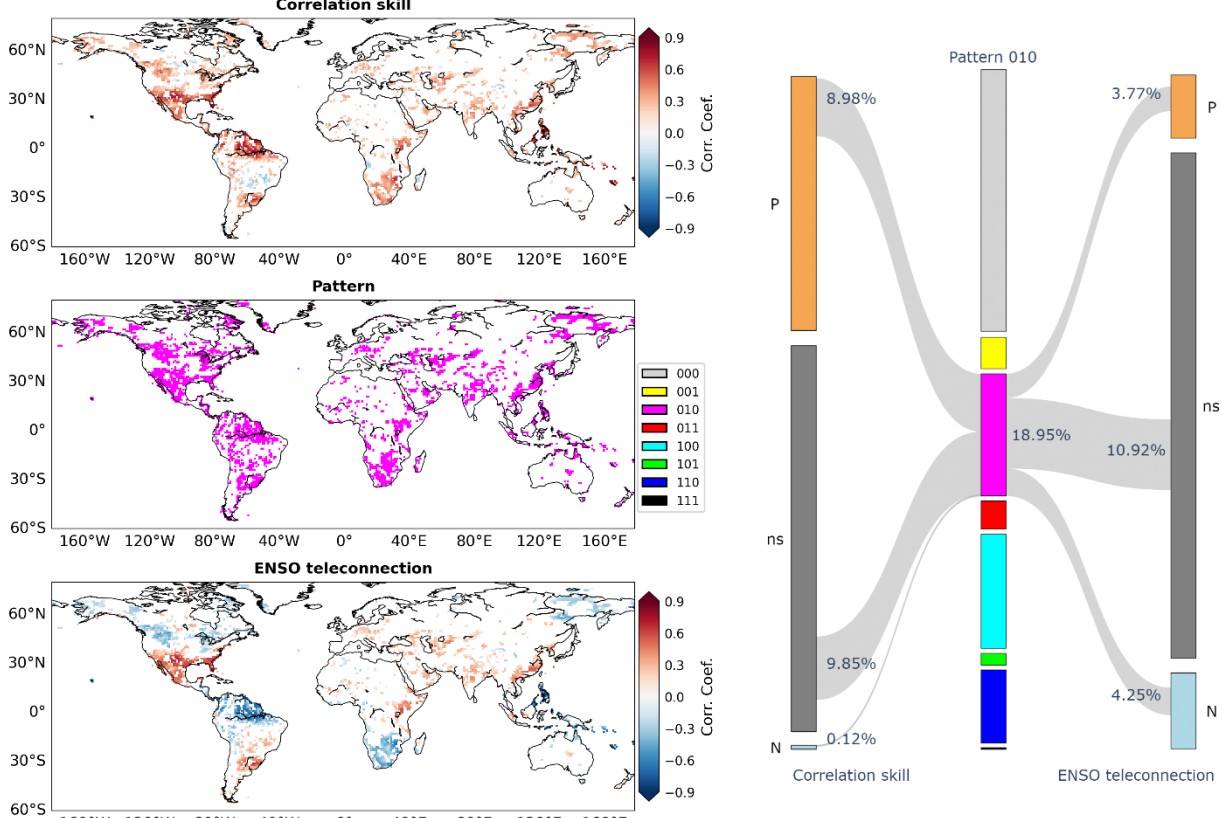


**Figure 4: Illustrations of correlation skill (upper left part) and ENSO teleconnection (lower left part) under the pattern 010 (middle left part) and Sankey diagram showing the percentages of grid cells exhibiting significantly positive (P), non-significant (ns) and significantly negative (N) correlation skill/ENSO teleconnection (right part). Grid cells under the other patterns are masked and therefore not shown in the spatial plots and the Sankey diagram.**


The pattern 100 focuses on the significant differing information of global precipitation in GCM forecasts from *Niño*3.4 index. From the left-hand side of Figure 5, it can be observed that this pattern (middle part in the left-hand side) tends to cover grid cells where correlation skill is around or above 0.3 (upper part in the left-hand side) but ENSO teleconnection is nearly zero (lower part in the left-hand side). This observation is confirmed by the right-hand side of Figure 5. As can be seen, while the

percentage of grid cells falling into the pattern 100 is 17.71%, the majority of them are with significantly positive correlation skill (15.76% in 17.71%) but all of them exhibit non-significant ENSO teleconnection (17.71% in 17.71%). These grid cells tend to locate in Europe and North Asia, where the influence of ENSO is limited and skillful GCM forecasts can relate to other teleconnections such as Arctic Oscillation and North Atlantic Oscillation (Hamouda et al., 2021).



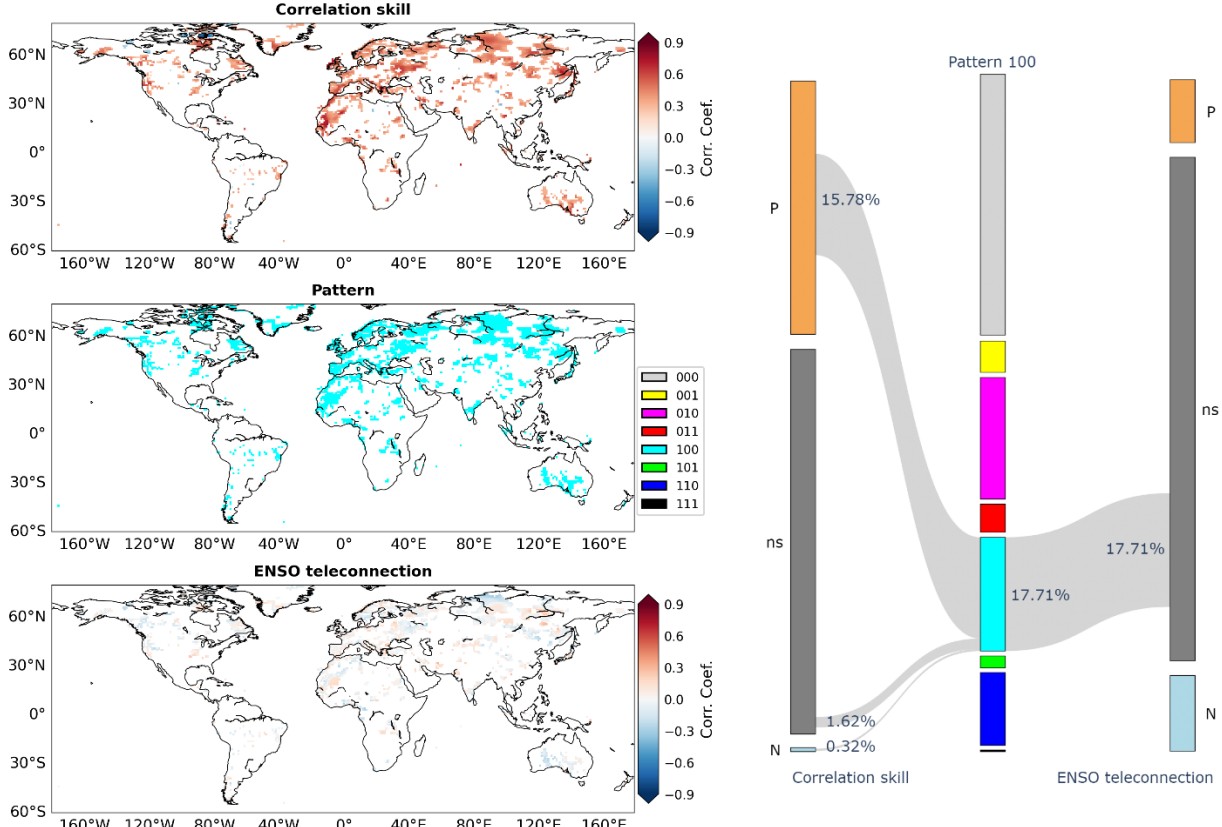

**Figure 5: As for Figure 4 but for the pattern 100.**

The pattern 110 indicates that the overlapping information is significant and that the differing information in GCM forecasts from *Niño*3.4 index is also significant. The implication is that regarding global seasonal precipitation in DJF, GCM forecasts not only contain information that is contained in *Niño*3.4 index but also provide a considerable amount of new information. At the left-hand side of Figure 6, some grid cells with pattern 110 are observed in southeast Australia, eastern Africa and northeastern Asia. Comparing Figure 6 to Figure 4, it is observed that some grid cells in southern North America, northern South America and southern Africa are under pattern 110, though many of them tend to be under pattern 100. Around the world, the percentage of grid cells falling into pattern 110 is 11.35%. For these grid cells, correlation skill is predominantly significantly positive (11.25% in 11.35%); in the meantime, ENSO teleconnection tends to be non-significant (7.09% in 11.35%).





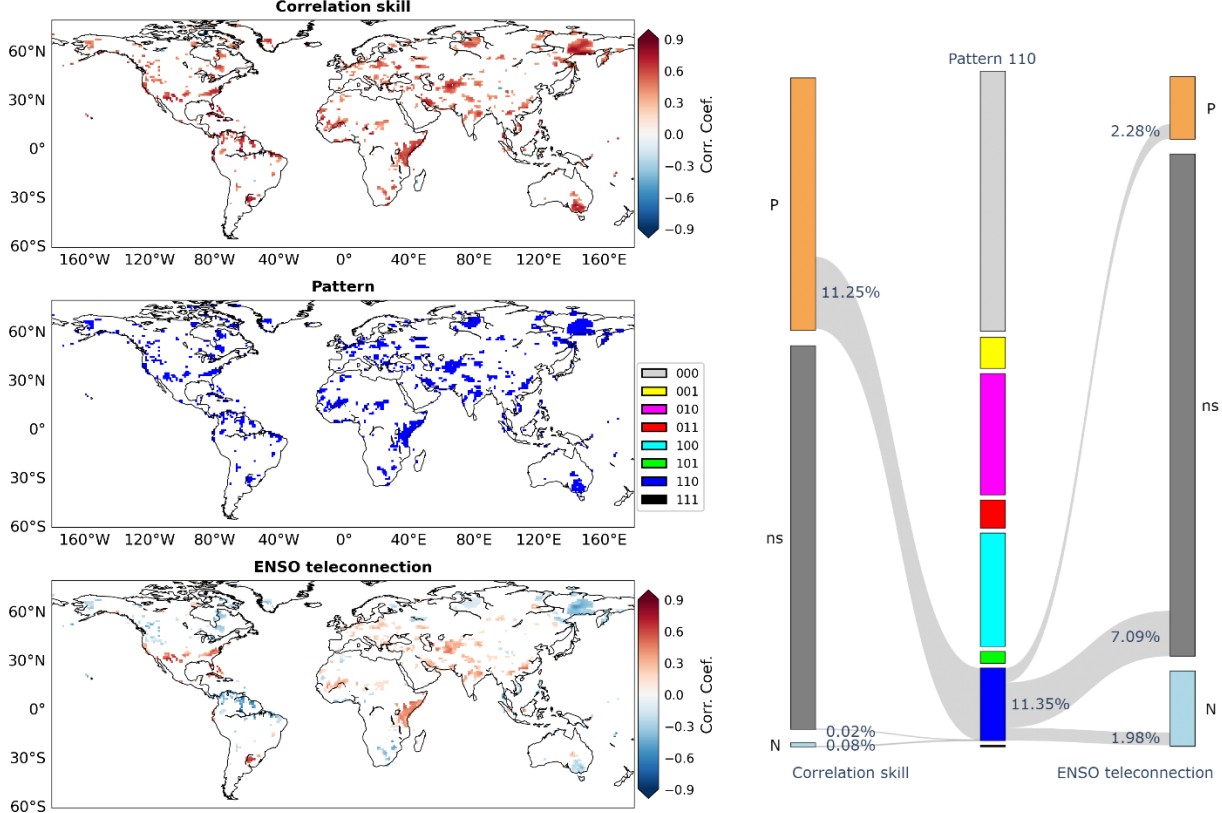

**Figure 6: As for Figure 4 but for the pattern 110.**


The pattern 001 pays attention to the differing information in *Niño*3.4 index from GCM forecasts. As shown in Figure 7, this pattern covers 4.87% of grid cells around the world. At the left-hand side of Figure 7, it is worthwhile to note that a number of grid cells in Western Australia exhibit significantly negative ENSO teleconnection but non-significant correlation skill. The implication is that therein GCM forecasts might have failed to account for the information of ENSO teleconnection. At the

right-hand side of Figure 7, it is observed that the majority of grid cells under pattern 001 are with neutral correlation skill (4.86% in 4.87%) and that their corresponding ENSO teleconnection can be significantly negative (2.21% in 4.87%) or significantly positive (1.73% in 4.87%).





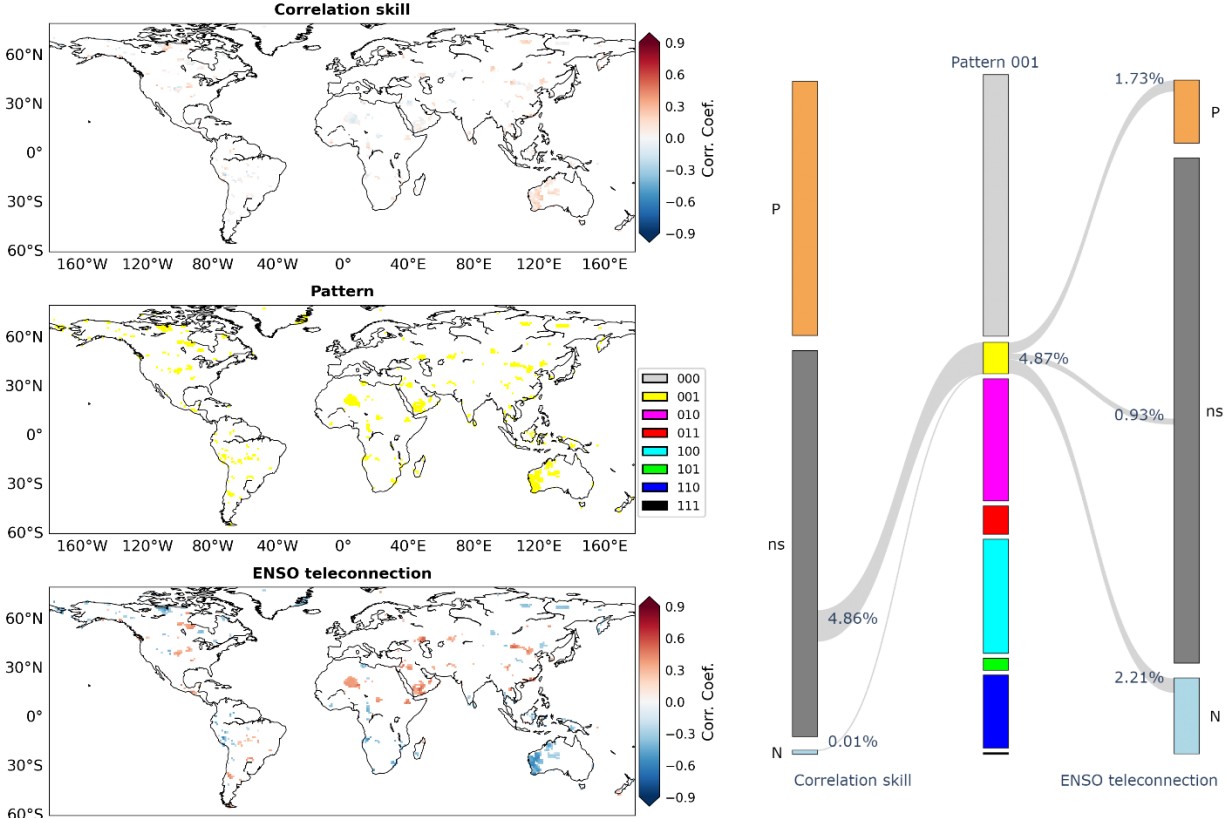

Figure 7: As for Figure 4 but for the pattern 001.

The pattern 011 indicates both significant overlapping information and significant differing information in *Niño*3.4 index from GCM forecasts. Grid cells exhibiting this pattern tend to be scattered in parts of southern North America, northern South America, Southeast Asia and southern Africa. They account for 4.38% of grid cells around the world. Among them, 1.72% exhibit significantly positive ENSO teleconnection and 2.66% significantly negative ENSO teleconnection. For these areas, the significant overlap suggests that a substantial amount of information in seasonal precipitation can be explained by both GCM forecasts and *Niño*3.4, while the significant differing information indicates the part that can only be explained by the *Niño*3.4 index.





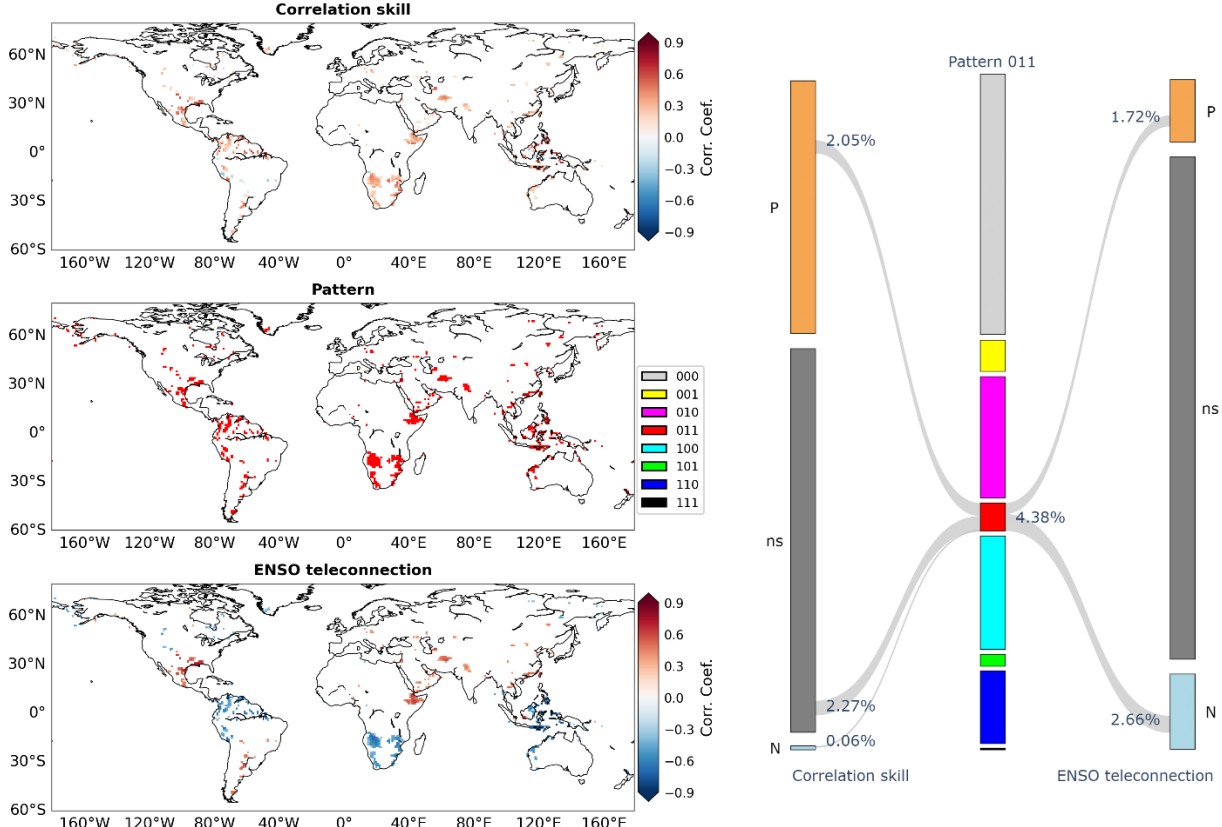

**Figure 8: As for Figure 4 but for the pattern 011.**

The patterns 101 is shown in Figure 9. It suggests that at some grid cells, the overlapping information is not significant but the two types of differing information are significant for both GCM forecasts and *Niño*3.4 index. About 1.86% of grid cells fall into this pattern.

The patterns 111 is shown in Figure 10. It implies that at some other grid cells, the overlapping information and the types of differing information can all be significant. It is noted that only 0.26% of grid cells around the world exhibit pattern 111.



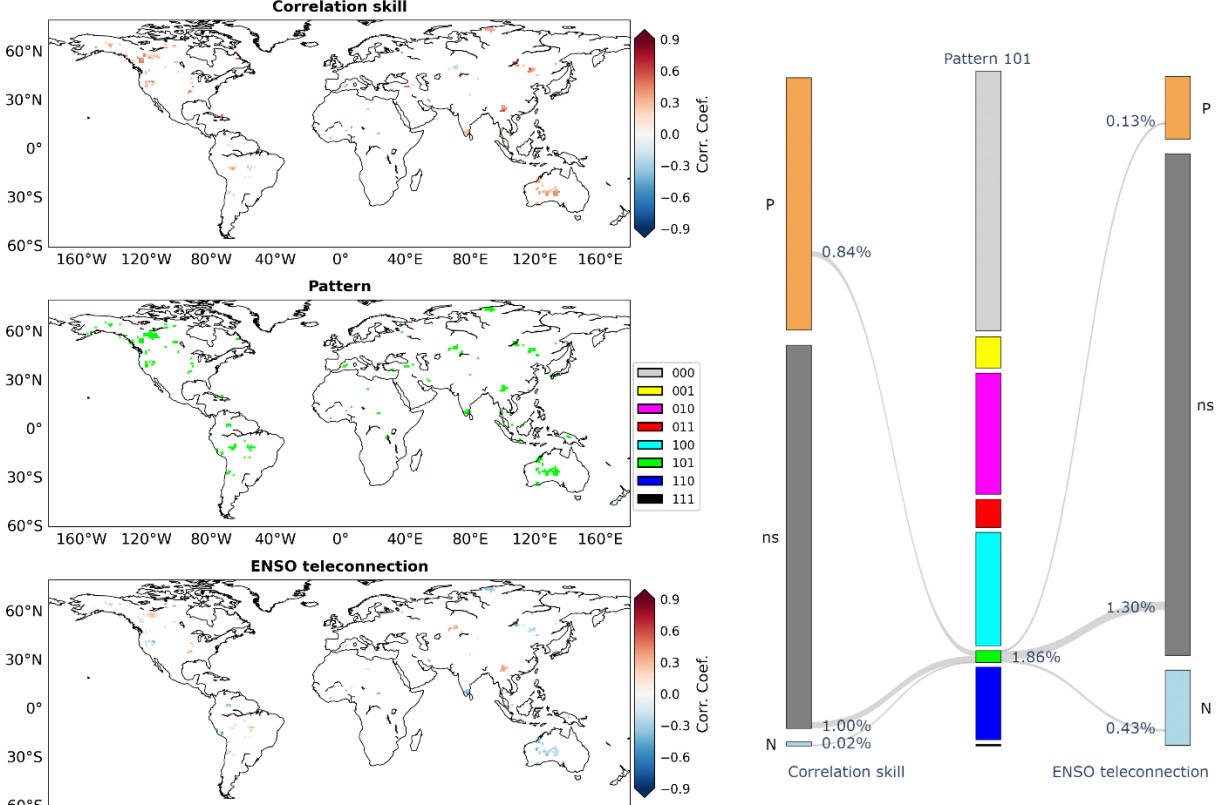

**Figure 9: As for Figure 4 but for the pattern 101.**




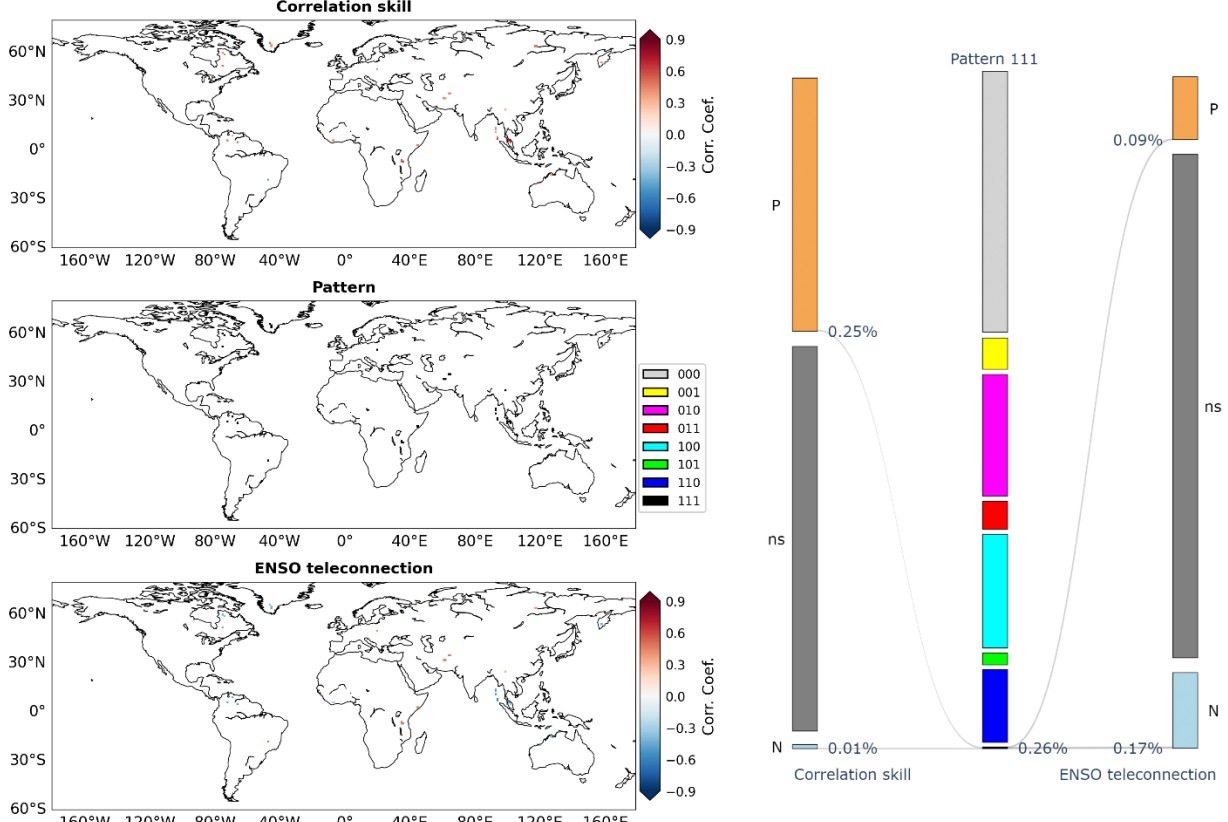

**Figure 10: As for Figure 4 but for the pattern 111.**

Among the eight patterns, the pattern 000 covers the most grid cells. At the left-hand side of Figure 11, it can be seen that grid
cells under the pattern 000 generally exhibit non-significant correlation skill and also non-significant ENSO teleconnection.
This result is in sharp contrast to pattern 010 that indicates reasonable correspondence between correlation skill and ENSO
teleconnection (Figure 4) and also to patterns 100 and 110 that suggest significantly positive correlation skill (Figures 5 and
6). Overall, the percentage of grid cells under pattern 000 is 40.62%. These grid cells predominantly exhibit neutral correlation
skill (40.30% in 40.62%) and neutral ENSO teleconnection (40.47% in 40.62%).




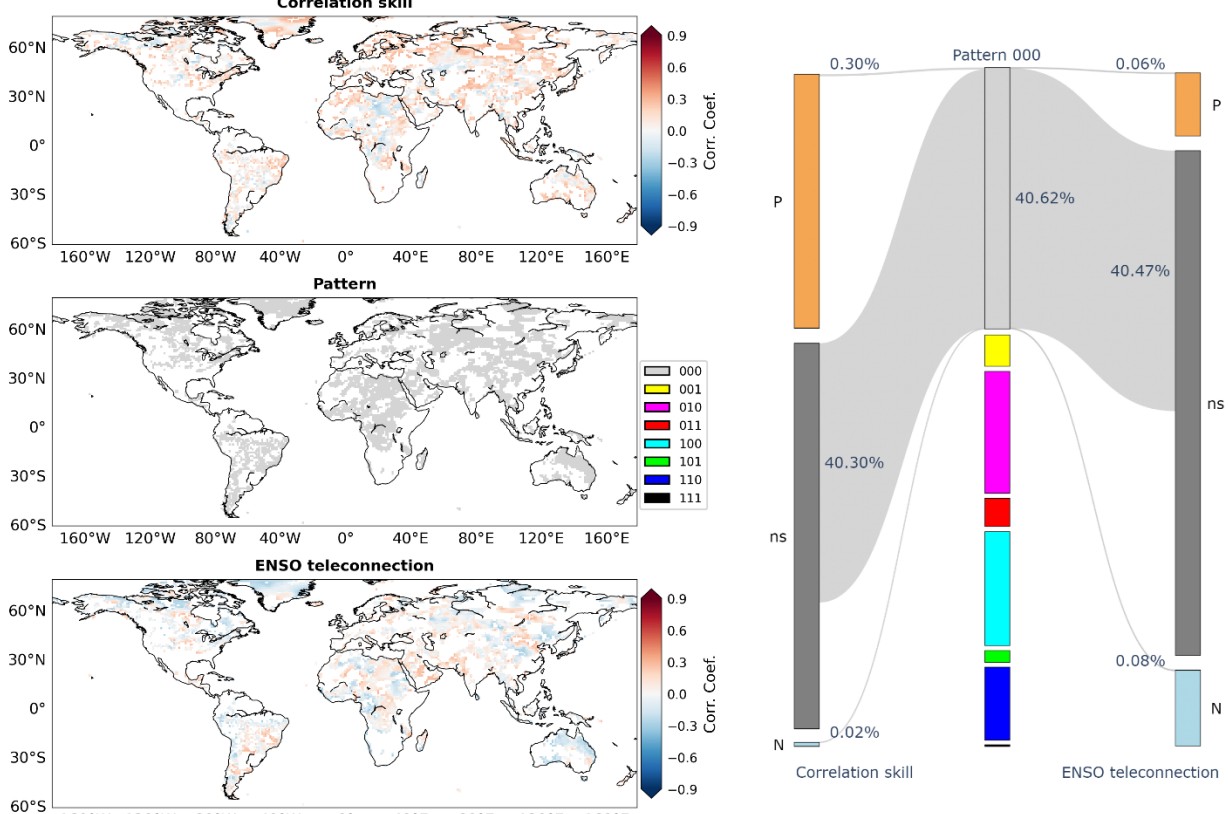

**Figure 11: As for Figure 4 but for the pattern 000.**

## 4.3 Association of correlation skill with ENSO teleconnection

GCM forecasts and Niño3.4 index generally represent two independent sources of information of global precipitation. For the patterns that indicate significant information, the percentage from the highest to the lowest is respectively 18.95% for pattern 010, 17.71% for pattern 100, 11.35% for pattern 110, 4.87% for pattern 001, 4.38% for pattern 011, 1.86% for pattern 101 and 0.26% for pattern 111. The GCM forecast correlation skill is plotted against the ENSO teleconnection by using scatter plots in Figure 12. Figure 12e in the center pools global land grid cells and employs the Viridis heatmap to indicate point density. It

can be observed that the correlation skill is largely positive and fall above the horizontal line. In addition, the heatmap suggests that the correlation skill tends to increase with the increased strength of ENSO teleconnection. These results suggest that the skill of GCM forecasts benefits from the prominence of ENSO teleconnection since GCMs tend to capture the influences of ENSO on the variability of global precipitation (Saha et al., 2014; Khan et al., 2017; Johnson et al., 2019; Becker et al., 2020; Delworth et al., 2020).





**Figure 12: Scatter plots of the association of GCM forecast correlation skill with ENSO teleconnection at the global scale with the heatmap indicating the density of scatter points (e). The relationship conditioned on the eight patterns are also illustrated, i.e., (a) pattern 000, (b) pattern 001, (c) pattern 010, (d) pattern 011, (f) pattern 100, (g) pattern 101, (h) pattern 110 and (i) pattern 111.**






Overall, the scatter plots under the eight patterns reveal a close but divergent association of correlation skill with ENSO teleconnection:

1) There exists significant overlapping information in GCM forecasts and *Niño*3.4 index under the patterns 010 (Figure 12c), 011 (Figure 12d), 110 (Figure 12h) and 111 (Figure 12i). The significance is for 34.94% of grid cells, i.e., 18.95% (010) + 4.38% (011) +11.35% (110) + 0.26% (111). From the corresponding scatter plots, it can be observed that both correlation skill and ENSO teleconnection ought to be reasonably high to facilitate significant overlapping information;

2) There is significant differing information in GCM forecasts from *Niño*3.4 index under the patterns 100 (Figure 12f), 101 (Figure 12g), 110 (Figure 12h) and 111 (Figure 12i). The significance is for 31.18% of global land grid cells, i.e., 17.71% (100) + 1.86% (101) + 11.35% (110) + 0.26% (111). Under these patterns, it is highlighted that the correlation skill tends to be higher than ENSO teleconnection. In particular, significantly positive correlation skill coincides with overall non-significant ENSO teleconnection under the pattern 100 in Figure 12f. Overall, these results imply that apart from ENSO, GCMs account for other hydro-climatic teleconnections to produce skilful precipitation forecasts (Saha et al., 2014; Johnson et al., 2019; Delworth et al., 2020);

3) There is significant differing information in *Niño*3.4 index from GCM forecasts under the patterns 001 (Figure 12b), 011 (Figure 12d), 101 (Figure 12g) and 111 (Figure 12i). The significance is for 11.37% of global land grid cells, i.e., 4.87% (001) + 4.38% (011) + 1.86% (101) + 0.26% (111). Under these patterns, ENSO teleconnection is generally higher than correlation skill. In particular, remarkable ENSO teleconnection coincide with overall non-significant correlation skill under the pattern 001 in Figure 12b. These results suggest that some ENSO teleconnection is still yet to be exploited by GCMs to improve precipitation forecast skill.

4) Neither the overlapping information nor the two types of differing information are significant under the pattern 000. It covers 40.62% of grid cells. From Figure 12a, it can be observed that either correlation skill or ENSO teleconnection is limited and that the corresponding scatter plot tends to cluster around the origin point. This result suggests that despite limited ENSO teleconnection, GCM forecasts still have plenty of room for improvement.

## 5 Discussion

The SOCD method is furthermore applied to investigate the effects of seasonality, lead time and significance level and also to elaborate on the patterns for the Indian Ocean Dipole (IOD, Cai et al., 2021). The additional results are presented in the supplementary material. 1) The effect of seasonality is shown in Figures S1 to S8. While regions exhibiting significant ENSO teleconnections vary from season to season (Figures S1 to S4), the eight patterns remain effective in characterizing the overlapping and differing information (Figures S1 to S8). 2) The effect of lead time is illustrated in Figures S9 to S12. At the lead times of 1 and 2 months, the percentage of the pattern 010 remains the highest among the seven patterns other than 000. This result highlights significant overlapping information in DJF, particularly over southern North America, northern South



America and Southern Africa. 3) The effect of significance level is shown in Figures S13 to S16. As the significant level is reduced from 0.10 to 0.05 and furthermore to 0.01, the percentage of the pattern 000 evidently increases but the seven patterns

that highlight significant overlapping and differing information remain. 4) The analysis is further extended to IOD in DJF (Figures S17 to S18). Comparing Figure S1 of ENSO to Figure 17 of IOD, the percentage of pattern 010 is reduced from 18.95% to 9.41% while the percentage of pattern 100 is increased from 17.71% to 22.83%. The indications are that IOD exhibits less overlapping information with GCM forecasts than ENSO does and that there is considerable differing information in GCM forecasts from IOD teleconnection.

Forecast skill is one of the most important issues for practical applications of dynamical global climate forecasts (Kirtman et al., 2014; Corti et al., 2015; Johnson et al., 2019). Among various measures of forecast skill, the correlation skill is probably the most popular owing to its simplicity in calculation and robustness to zero and missing values (Barnston et al., 2012; Ma et al., 2016; Slater et al., 2019). From spatial plots of correlation skill at regional or global scales, it can be observed where GCM forecasts are skilful and where GCM forecasts are not satisfactory (Ma et al., 2016; Slater et al., 2019; Delworth et al., 2020;

Medina and Tian, 2020). Previously, it was observed that GCM forecasts tend to be skilful in regions subject to the prominent influences of ENSO; accordingly, forecast skill is attributed to the effectiveness of GCMs in capturing ENSO-related climate dynamics (Kirtman et al., 2014; Slater et al., 2019; Lin et al., 2020). In this paper, the developed SOCD method not only confirms the significant overlapping information but also highlights that there is significant differing information in GCM forecasts from ENSO teleconnection for 31.18% of global land grid cells and that there is significant differing information in

ENSO teleconnection from GCM forecasts for 11.37% of grid cells. In the future, this method can be extended to other hydroclimatic teleconnections to explore the sources of predictability for GCM forecasts.

## 6 Conclusions

While ENSO teleconnection has been conventionally used in hydroclimatic forecasting of regional precipitation and

streamflow, GCM forecasts are increasingly available for hydrological applications. It is important to investigate to what extent emerging GCM forecasts provide "new" information compared to conventional ENSO teleconnection. The SOCD method developed in this paper addresses this issue through the mathematical formulation of set operations. Specifically, the union operation quantifies the information of global seasonal precipitation contained in both GCM forecasts and Niño3.4 index; the intersection operation derives the overlapping information of global precipitation in GCM forecasts and Niño3.4 index; and

furthermore, the difference operations illustrates two types of differing information, i.e., the differing information in GCM forecasts from Niño3.4 index and the differing information in Niño3.4 index from GCM forecasts. The significance tests of the three types of information facilitate in total eight patterns to disentangle the close but divergent association of GCM forecast correlation skill with ENSO teleconnection. GCM forecasts and Niño3.4 index are generally two independent sources of data for hydroclimatic forecasting. While the existence of significant overlapping information suggests that they can provide some

similar information, the existence of significant differing information indicates that the two data sources can also be complementary to each other. In the future, more efforts can be devoted to investigating more GCM forecast datasets and more hydroclimatic teleconnections to yield insights into the forecast skill of GCM forecasts and to facilitate applications of GCM forecasts to hydrological modelling and water resources management.

## Acknowledgments

This research is supported by the National Key Research and Development Program of China (2021YFC3001000), the National Natural Science Foundation of China (51979295, 51861125203, 52109046 and U1911204) and the Guangdong Provincial Department of Science and Technology (2019ZT08G090).

## Data Availability Statement

The forecast and observation datasets are downloaded from https://iridl.ldeo.columbia.edu/SOURCES/.Models/.NMME/. The Niño3.4 index is downloaded from https://www.cpc.ncep.noaa.gov/data/indices/.

## Author contribution

TZ and XC designed the experiments and HC carried them out. HC and YT developed the model code and performed the simulations. TZ prepared the manuscript with contributions from all co-authors.

## Competing interests

The authors declare that they have no conflict of interest.

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
