# Peer review of "Quantifying overlapping and differing information of global precipitation for GCM forecasts and El Niño–Southern Oscillation"

_Hydrology and Earth System Sciences, 2022_

## Author Comment (AC2)

**Response**

*Anonymous Referee #2*:

*This manuscript develops a novel Set Operations of Coefficients of Determination (SOCD) method to explicitly quantify the overlapping and differing information for GCM forecasts and ENSO teleconnection. The proposed method and its case study are interesting and well presented.*

We are grateful to you for the positive comments.

*I have some comments, especially concerning the conclusions derived from the case study.*

Thank you very much for the constructive comments. We have improved the paper accordingly and provide point-by-point responses.

*This study derives the patterns from CFSv2 forecasts and observations. However, we should note that the forecast skills of GCMs are of high uncertainty in different models. The ability of GCMs in capturing ENSO-related climate dynamics are different. Therefore, the results and conclusions may be different when other GCMs are used. The authors should have a more detailed discussion on this issue.*

Thank you for the insightful comment. We have devised an additional experiment to demonstrate the different ability of GCMs in capturing ENSO teleconnections and synthesize the results in the Discussion section:

"The SOCD method is also extended to evaluate the overlapping and differing information under other GCM forecasts and hydroclimatic teleconnections. In the supplementary material, Figures S19 and S20 show the results for the CanCM4 forecasts generated at the Canadian Meteorological Center (CMC) (Merryfield et al., 2013). It can be observed that the percentage of the pattern 000 is higher than that for CFSv2 forecasts. The CanCM4 forecasts seem to be less skilful in Europe but more skilful in the western part of Australia. These results suggest that different GCM forecasts can be complementary to each other in different regions and that they can be combined to generate more skilful forecasts (Kirtman et al., 2014; Slater et al., 2019; Schepen et al., 2020)." (Page 23, Lines 369 to 375)

[Figure]

Figure S19. As for Figure 12, but for CMC2-CanCM4 forecasts in DJF

[Figure]

Figure S20. As for Figure 13, but for CMC2-CanCM4 forecasts in DJF

*Another issue is that the lagged relationship between ENSO and seasonal precipitation that always matters. The lagged climate indices have been widely used as predictors in previous studies. In this study, the concurrent relationship between ENSO and seasonal precipitation is analyzed. We suggest that the lagged relationship should also be discussed.*

Thank you for the constructive comment. An additional experiment is conducted to investigate the effects of lag time:

"The SOCD method is furthermore applied to investigate the eight patterns considering the effects of seasonality, lead time, lag time and significance level. The additional results are presented in the supplementary material. … 3) The effect of the lag time of Niño3.4 index is illustrated in Figures S11 to S14. Compared to the concurrent

teleconnection, the spatial distribution of the eight patterns tends to be similar for monthly Niño3.4 index at the lag times of 1 and 2 months, with a slight increase in the percentage of the pattern 000. The result confirm the temporal persistency in the Niño3.4 index (Yang et al., 2018). …" (Pages 22 to 23, Lines 358 to 370)

[Figure]

Figure S11. As for Figure 12, but for monthly Niño3.4 index at the lag time of 1 month

[Figure]

Figure S12. As for Figure 12, but for monthly Niño3.4 index at the lag time of 2 months

[Figure]

Figure S13. As for Figure 13, but for monthly Niño3.4 index at the lag time of 1 month

[Figure]

Figure S14. As for Figure 13, but for monthly Niño3.4 index at the lag time of 2 months

*The SOCD method uses three classic simple linear regression models to account for the information of observations in forecasts and nino3.4 index. However, this assumption may not be sufficient. The linear regression may lose some important information, especially for extreme seasonal precipitation events. This should be discussed in the discussion section as well.*

Thank you for the insightful comment. The correlation skill that assumes a linear relationship and the existence of possible nonlinear relationships are illustrated in the discussion section:

"The correlation skill is one of the most popular measures of forecast skill owing to its simplicity in calculation and robustness to zero and missing values (Barnston et al., 2012; Yuan et al., 2014; Ma et al., 2016; Slater et al., 2019; Huang and Zhao, 2022).

From spatial plots of correlation skill at regional or global scales, it can be observed where GCM forecasts are skilful and where GCM forecasts are not satisfactory (Ma et al., 2016; Slater et al., 2019; Delworth et al., 2020). Previously, it was observed that GCM forecasts tend to be skilful in regions subject to prominent influences of ENSO; accordingly, forecast skill is attributed to the effectiveness of GCMs in capturing ENSO-related climate dynamics (Kirtman et al., 2014; Slater et al., 2019; Lin et al., 2020). In this paper, the developed SOCD method not only confirms the significant overlapping information but also highlights that there exists significant differing information in GCM forecasts from ENSO teleconnection for 31.18% of global land grid cells and that there is significant differing information in ENSO teleconnection from GCM forecasts for 11.37% of grid cells. It is noted that the simple linear regression only accounts for linear relationships. Possible nonlinear relationships between forecasts and observations suggest the usage of nonlinear models in future analysis of the overlapping and differing information (Strazzo et al., 2019; Schepen et al., 2020; Li et al., 2021)." (Page 23, Lines 381 to 392)

---

## Author Response (AR1)

**Response**

*Editor decision:*

*We kindly ask you to revise your manuscript accordingly and to upload the revised files, a point-by-point reply to the comments, and a marked-up manuscript version showing the changes made no later than 04 Jul 2022 at: https://editor.copernicus.org/HESS/review-file-upload/hess-2022-58*

We are grateful to you for the kind decision. We have conducted a thorough revision to improve the paper by following the insightful and constructive comments provided by the reviewers. Plase see the point-by-point responses in the following.

**Response**

*Anonymous Referee #1:*

*This is an excellent and interesting study. The authors have adequately addressed all the comments raised by previous reviewers.*

Thank you very much for the positive comments.

*Just one minor point. I think in the Introduction, the authors should appreciate the latest advances in the seasonal hydroclimate forecast using hybrid dynamic-statistical approaches, such as Wanders et al. (2017). Seasonal forecast is also key for drought impact reduction, e.g., related to food security and water resources management (He et al., 2019; Sheffield et al., 2014; He et al., 2021).*

*Ref:*

*He, X., Estes, L., Konar, M., Tian, D., Anghileri, D., Baylis, K., Evans, T.P. and Sheffield, J., 2019. Integrated approaches to understanding and reducing drought impact on food security across scales. Current Opinion in Environmental Sustainability, 40, pp.43-54.*

*Wanders, N., Bachas, A., He, X.G., Huang, H., Koppa, A., Mekonnen, Z.T., Pagán, B.R., Peng, L.Q., Vergopolan, N., Wang, K.J. and Xiao, M., 2017. Forecasting the hydroclimatic signature of the 2015/16 El Niño event on the Western United States. Journal of Hydrometeorology, 18(1), pp.177-186.*

*Sheffield, J., Wood, E.F., Chaney, N., Guan, K., Sadri, S., Yuan, X., Olang, L., Amani, A., Ali, A., Demuth, S. and Ogallo, L., 2014. A drought monitoring and forecasting system for sub-Sahara African water resources and food security. Bulletin of the American Meteorological Society, 95(6), pp.861-882.*

*He, X., Bryant, B.P., Moran, T., Mach, K.J., Wei, Z. and Freyberg, D.L., 2021. Climate-informed hydrologic modeling and policy typology to guide managed aquifer recharge. Science advances, 7(17), p.eabe6025.*

Thank you for the constructive comment. We have incorporated the important references into the Introduction:

"Seasonal hydroclimatic forecasts are important for agricultural scheduling, water management and drought mitigation (Sheffield et al., 2014; Anghileri et al., 2016; Zhao

et al., 2019; Peng et al., 2018; He et al., 2019). Performing hydroclimatic forecasting into the future, the uncertainty generally arises from catchment initial conditions and future climate forcings (Wood and Lettenmaier, 2006; Huang et al., 2020; Yuan et al., 2014). In a short lead time up to about one month, initial conditions tend to outweigh climate forcings; at longer lead times, climate forcings become a more important contributor (Li et al., 2009; Yossef et al., 2013). Therefore, besides remote sensing-based estimations of initial conditions of snow cover, soil moisture and groundwater storage (Mei et al., 2020; Xu et al., 2020b; Sheffield et al., 2014), efforts have been devoted to developing sub-seasonal to seasonal hydroclimatic forecasts of temperature and precipitation (Schepen et al., 2020; Strazzo et al., 2019; Bennett et al., 2016; Cash et al., 2019; Li et al., 2017). While temperature forecasts have been improved substantially in the past decades, the generation of skilful precipitation forecasts remains a challenging task (Becker et al., 2022)." (Pages 1 to 2, Lines 27 to 36)

"Conventional ENSO-based statistical forecasts and emerging GCM dynamical forecasts generally represent two different sources of information (Wood and Lettenmaier, 2006; Bauer et al., 2015; Emerton et al., 2017; Delworth et al., 2020; He et al., 2021). While both of them are valuable and they can further be combined to generate improved forecasts (Madadgar et al., 2016; Wanders et al., 2017; Strazzo et al., 2019), it is not yet known to what extent their information overlaps or differs. Small overlap and large difference highlight that GCM forecasts do offer new information comparing to ENSO teleconnection, while large overlap and small difference imply that GCM forecasts might not provide additional information. Zhao et al. (2021) investigated the overlapping information to attribute GCM forecast correlation skill to ENSO teleconnection. In this paper, we build a Set Operations of Coefficients of Determination (SOCD) method upon Zhao et al. (2021) to furthermore account for the differing information. As will be demonstrated through the methods and results, besides the overlapping information, there exist two types of differing information, i.e., the differing information in GCM forecasts from ENSO and the differing information in ENSO from GCM forecasts. The three types of information facilitate eight patterns to disentangle the close but divergent association of GCM correlation skill with ENSO teleconnection." (Pages 2 to 3, Lines 58 to 69)

Anghileri, D., Voisin, N., Castelletti, A., Pianosi, F., Nijssen, B., & Lettenmaier, D. P.: Value of long-term streamflow forecasts to reservoir operations for water supply in snow-dominated river catchments, Water Resources Research, 52(6), 4209-4225, https://doi.org/10.1002/2015wr017864, 2016.

He, X., Bryant, B.P., Moran, T., Mach, K.J., Wei, Z. and Freyberg, D.L.: Climate-informed hydrologic modeling and policy typology to guide managed aquifer recharge,

Science advances, 7(17), eabe6025, https://doi.org/10.1126/sciadv.abe6025, 2021.

He, X., Estes, L., Konar, M., Tian, D., Anghileri, D., Baylis, K., Evans, T.P. and Sheffield, J.: Integrated approaches to understanding and reducing drought impact on food security across scales, Current Opinion in Environmental Sustainability, 40, 43-54, https://doi.org/10.1016/j.cosust.2019.09.006, 2019.

Madadgar, S., AghaKouchak, A., Shukla, S., Wood, A. W., Cheng, L., Hsu, K.-L., and Svoboda, M.: A hybrid statistical-dynamical framework for meteorological drought prediction: Application to the southwestern United States, Water Resources Research, 52, 5095– 5110, https://doi.org/10.1002/2015WR018547, 2016.

Peng, B., Guan, K., Pan, M., & Li, Y.: Benefits of Seasonal Climate Prediction and Satellite Data for Forecasting U.S. Maize Yield. Geophysical Research Letters, 45(18), 9662-9671. https://doi.org/10.1029/2018gl079291, 2018

Sheffield, J., Wood, E.F., Chaney, N., Guan, K., Sadri, S., Yuan, X., Olang, L., Amani, A., Ali, A., Demuth, S. and Ogallo, L., 2014. A drought monitoring and forecasting system for sub-Sahara African water resources and food security. Bulletin of the American Meteorological Society, 95(6), pp.861-882. https://doi.org/10.1175/BAMS-D-12-00124.1

Wanders, N., Bachas, A., He, X.G., Huang, H., Koppa, A., Mekonnen, Z.T., Pagán, B.R., Peng, L.Q., Vergopolan, N., Wang, K.J. and Xiao, M.: Forecasting the hydroclimatic signature of the 2015/16 El Niño event on the Western United States, Journal of Hydrometeorology, 18(1), 177-186, https://doi.org/10.1175/JHM-D-16-0230.1, 2017

**Response**

*Anonymous Referee #2*:

*This manuscript develops a novel Set Operations of Coefficients of Determination (SOCD) method to explicitly quantify the overlapping and differing information for GCM forecasts and ENSO teleconnection. The proposed method and its case study are interesting and well presented. I have some comments, especially concerning the conclusions derived from the case study.*

We are grateful to you for the constructive comments. We have improved the paper accordingly and provide point-by-point responses.

*This study derives the patterns from CFSv2 forecasts and observations. However, we should note that the forecast skills of GCMs are of high uncertainty in different models. The ability of GCMs in capturing ENSO-related climate dynamics are different. Therefore, the results and conclusions may be different when other GCMs are used. The authors should have a more detailed discussion on this issue.*

Thank you for the insightful comment. We have devised an additional experiment to demonstrate the different ability of GCMs in capturing ENSO teleconnections and synthesize the results in the Discussion section:

"The SOCD method is also extended to evaluate the overlapping and differing information under other GCM forecasts and hydroclimatic teleconnections. In the supplementary material, Figures S19 and S20 show the results for the CanCM4 forecasts generated at the Canadian Meteorological Center (CMC) (Merryfield et al., 2013). It can be observed that the percentage of the pattern 000 is higher than that for CFSv2 forecasts. The CanCM4 forecasts seem to be less skilful in Europe but more skilful in the western part of Australia. These results suggest that different GCM forecasts can be complementary to each other in different regions and that they can be combined to generate more skilful forecasts (Kirtman et al., 2014; Slater et al., 2019; Schepen et al., 2020)." (Page 23, Lines 370 to 376)

[Figure]

Figure S19. As for Figure 12, but for CMC2-CanCM4 forecasts in DJF

[Figure]

Figure S20. As for Figure 13, but for CMC2-CanCM4 forecasts in DJF

*Another issue is that the lagged relationship between ENSO and seasonal precipitation that always matters. The lagged climate indices have been widely used as predictors in previous studies. In this study, the concurrent relationship between ENSO and seasonal precipitation is analyzed. We suggest that the lagged relationship should also be discussed.*

Thank you for the comment. A new experiment is performed for monthly Niño3.4 index at the lag time of 1 and 2 months and the results are presented in Supplementary Material:

"The SOCD method is furthermore applied to investigate the eight patterns considering the effects of seasonality, lead time, lag time and significance level. The additional results are presented in the supplementary material. 1) The effect of

seasonality is shown in Figures S1 to S6. It can be observed that regions exhibiting significant ENSO teleconnections vary by season (Figures S1 to S3) and that the eight patterns remain effective in characterizing the overlapping and differing information (Figures S4 to S6). 2) The effect of lead time is illustrated in Figures S7 to S10. At the lead times of 1 and 2 months, the percentage of the pattern 010 remains the highest among the seven patterns other than 000. This result highlights significant overlapping information in DJF, particularly over southern North America, northern South America and Southern Africa. 3) The effect of the lag time of Niño3.4 index is illustrated in Figures S11 to S14. Compared to concurrent teleconnection, for monthly Niño3.4 index at the lag times of 1 and 2 months, the spatial distribution of the eight patterns tends to be similar with a slightly increasing in the percentage of the pattern 000. The results confirm the temporal persistency in the Niño3.4 index (Yang et al., 2018). 4) The effect of the significance level is shown in Figures S15 to S18. As the significance level is reduced from 0.10 to 0.05 and furthermore to 0.01, the percentage of the pattern 000 evidently increases but the seven patterns that highlight significant overlapping and differing information remain." (Pages 22 to 23, Lines 357 to 369)

[Figure]

Figure S11. As for Figure 12, but for monthly Niño3.4 index at the lag time of 1 month

[Figure]

Figure S12. As for Figure 12, but for monthly Niño3.4 index at the lag time of 2 months

[Figure]

Figure S13. As for Figure 13, but for monthly Niño3.4 index at the lag time of 1 month

[Figure]

Figure S14. As for Figure 13, but for monthly Niño3.4 index at the lag time of 2 months

*The SOCD method uses three classic simple linear regression models to account for the information of observations in forecasts and nino3.4 index. However, this assumption may not be sufficient. The linear regression may lose some important information, especially for extreme seasonal precipitation events. This should be discussed in the discussion section as well.*

Thank you. In the revision, the linear assumption has been noted in the Discussion section:

"The correlation skill is one of the most popular measures of forecast skill owing to its simplicity in calculation and robustness to zero and missing values (Barnston et al., 2012; Ma et al., 2016; Slater et al., 2019; Huang and Zhao, 2022; Yuan et al., 2014). From spatial plots of correlation skill at regional or global scales, it can be observed

where GCM forecasts are skilful and where GCM forecasts are not satisfactory (Ma et al., 2016; Slater et al., 2019; Delworth et al., 2020). Previously, it was observed that GCM forecasts tend to be skilful in regions subject to prominent influences of ENSO; accordingly, forecast skill is attributed to the effectiveness of GCMs in capturing ENSO-related climate dynamics (Kirtman et al., 2014; Slater et al., 2019; Lin et al., 2020). In this paper, the developed SOCD method not only confirms the significant overlapping information but also highlights that there exists significant differing information in GCM forecasts from ENSO teleconnection for 31.18% of global land grid cells and that there is significant differing information in ENSO teleconnection from GCM forecasts for 11.37% of grid cells. It is noted that the simple linear regression only accounts for linear relationships and that the possible nonlinear relationships between forecasts and observations suggest the usage of nonlinear models in future analysis of the overlapping and differing information (Li et al., 2021; Schepen et al., 2020; Strazzo et al., 2019)." (Page 23, Lines 380 to 391)

---

## Referee Report (RR1)

I have reviewed this second submission and am pleased to see the authors' response. Most of my comments are either editorial or require clarifications.

L. 11: "hydroclimatic forecasting" -> "precipitation forecasting"

I think that the word "hydroclimatic forecasting" is beyond the scope of this study. The authors should focus on precipitation forecasts. Same remark at L. 27, L. 28, et al.

L. 36: " skilful precipitation forecasts remain  challenging

L. 75: "start"-> "initial", same remark at L. 75, L. 81, L. 101, L. 103, L. 107.

L. 92: "monthly Nino3.4 is" -> "monthly Nino3.4 index is"

L. 328:  other eight subplots

L. 433: Journal title is missing, Same remark at L. 437, L. 453, L. 506, L. 508, L. 524, L. 533, L. 535, L. 544, L. 547, L. 557, L. 570, L. 572.

---

## Editor Decision (ED1)

Hydrol. Earth Syst. Sci. Discuss., referee comment RC1 https://doi.org/10.5194/hess-2022-58-RC1, 2022 © Author(s) 2022. This work is distributed under the Creative Commons Attribution 4.0 License.

**Comment on hess-2022-58**

Anonymous Referee #1

Referee comment on "Quantifying overlapping and differing information of global precipitation for GCM forecasts and El Niño–Southern Oscillation" by Tongtiegang Zhao et al., Hydrol. Earth Syst. Sci. Discuss., https://doi.org/10.5194/hess-2022-58-RC1, 2022

This is an excellent and interesting study. The authors have adequately addressed all the comments raised by previous reviewers.

Just one minor point. I think in the Introduction, the authors should appreciate the latest advances in the seasonal hydroclimate forecast using hybrid dynamic-statistical approaches, such as Wanders et al. (2017). Seasonal forecast is also key for drought impact reduction, e.g., related to food security and water resources management (He et al., 2019; Sheffield et al., 2014; He et al., 2021).

Ref:

He, X., Estes, L., Konar, M., Tian, D., Anghileri, D., Baylis, K., Evans, T.P. and Sheffield, J., 2019. Integrated approaches to understanding and reducing drought impact on food security across scales. Current Opinion in Environmental Sustainability, 40, pp.43-54.

Wanders, N., Bachas, A., He, X.G., Huang, H., Koppa, A., Mekonnen, Z.T., Pagán, B.R., Peng, L.Q., Vergopolan, N., Wang, K.J. and Xiao, M., 2017. Forecasting the hydroclimatic signature of the 2015/16 El Niño event on the Western United States. Journal of Hydrometeorology, 18(1), pp.177-186.

Sheffield, J., Wood, E.F., Chaney, N., Guan, K., Sadri, S., Yuan, X., Olang, L., Amani, A., Ali, A., Demuth, S. and Ogallo, L., 2014. A drought monitoring and forecasting system for sub-Sahara African water resources and food security. Bulletin of the American Meteorological Society, 95(6), pp.861-882.

He, X., Bryant, B.P., Moran, T., Mach, K.J., Wei, Z. and Freyberg, D.L., 2021. Climateinformed hydrologic modeling and policy typology to guide managed aquifer recharge. Science advances, 7(17), p.eabe6025.

Hydrol. Earth Syst. Sci. Discuss., referee comment RC2 https://doi.org/10.5194/hess-2022-58-RC2, 2022 © Author(s) 2022. This work is distributed under the Creative Commons Attribution 4.0 License.

**Comment on hess-2022-58**

Anonymous Referee #2

Referee comment on "Quantifying overlapping and differing information of global precipitation for GCM forecasts and El Niño–Southern Oscillation" by Tongtiegang Zhao et al., Hydrol. Earth Syst. Sci. Discuss., https://doi.org/10.5194/hess-2022-58-RC2, 2022

This manuscript develops a novel Set Operations of Coefficients of Determination (SOCD) method to explicitly quantify the overlapping and differing information for GCM forecasts and ENSO teleconnection. The proposed method and its case study are interesting and well presented. I have some comments, especially concerning the conclusions derived from the case study.

- This study derives the patterns from CFSv2 forecasts and observations. However, we should note that the forecast skills of GCMs are of high uncertainty in different models. The ability of GCMs in capturing ENSO-related climate dynamics are different. Therefore, the results and conclusions may be different when other GCMs are used. The authors should have a more detailed discussion on this issue.
- Another issue is that the lagged relationship between ENSO and seasonal precipitation that always matters. The lagged climate indices have been widely used as predictors in previous studies. In this study, the concurrent relationship between ENSO and seasonal precipitation is analyzed. We suggest that the lagged relationship should also be discussed.
- The SOCD method uses three classic simple linear regression models to account for the information of observations in forecasts and nino3.4 index. However, this assumption may not be sufficient. The linear regression may lose some important information, especially for extreme seasonal precipitation events. This should be discussed in the discussion section as well.